# High-Temperature Corrosion of Ni-Cr-Mo Cladding Layers with Different Si Contents in NaCl-KCl-Na_2_SO_4_-K_2_SO_4_ Mixed Salt Medium

**DOI:** 10.3390/ma15093152

**Published:** 2022-04-27

**Authors:** Shanshan Chen, Zongde Liu, Yue Shen, Simin Liu

**Affiliations:** Key Laboratory of Energy Transfer and System of Power Station of Ministry of Education, North China Electric Power University, Beijing 102206, China; cjnscss@163.com (S.C.); syy@ncepu.edu.cn (Y.S.); liusimin1016@163.com (S.L.)

**Keywords:** high-temperature corrosion, laser cladding, Si content, Ni-Cr-Mo

## Abstract

In this work, Ni-Cr-Mo cladding layers with different Si contents were prepared on Q235 steel using laser-cladding technology, and their corrosion characteristics were investigated in NaCl-KCl-Na_2_SO_4_-K_2_SO_4_ mixed salt at 550 °C. The corrosion resistance of each cladding layer was tested by weight loss method, and the phase compositions and microstructures of the cladding layers and corrosion products were determined by X-ray diffraction (XRD) and scanning electron microscopy (SEM). The results show that Si contributed to the formation of a dense chromium oxide film on the surface, and the addition of Si can significantly improve the corrosion resistance of the cladding layer at high temperature. At 550 °C, the corrosion rate of the cladding layer with 5 wt.% Si was only 38.2% of that of the cladding layer without Si. After 168 h of high-temperature corrosion, no Cr-rich oxide scale was found in the outermost layer of the Ni-Cr-Mo cladding layer without Si. When Si content was 3 wt.% and 5 wt.%, the Cr-rich oxide scale of the cladding layer was denser than that of the coating with 1 wt.% Si content.

## 1. Introduction

With the production of more and more urban household garbage, garbage incineration is often adopted as a means of avoiding pollution problems [1,2]. Garbage-incineration power generation can facilitate the recycling and utilization of garbage classification resources and can also achieve the purpose of environmental protection. Unfortunately, waste composition is very complicated, with high organic content and harmful substances, such as Cl, S and alkali metals. These components can generate chemicals, such as HCl, SO_2_, H_2_S, chlorine salts (NaCl, KCl) and sulfates (such as Na_2_SO_4_, K_2_SO_4_), after high-temperature incineration. Thus, waste-incineration power generation boilers suffer from chloride-type high-temperature corrosion and sulfide-type high-temperature corrosion [3,4]. High-temperature corrosion limits the promotion and development of waste-incineration technology [5]. In a waste-incineration power generation boiler affected by high temperature and combustion-waste smoke corrosion due to the harsh service conditions and complex environment, the heating surface tubes (water wall tube, superheater tube, reheat tube, economizer tube) will be subjected to high-temperature oxidation, vulcanization and chlorine corrosion, which can easily cause tube explosion accident and threaten the safe operation of the generator set. Therefore, it is of great practical significance to solve the high-temperature corrosion problem of waste-incineration boilers.

Due to their excellent corrosion resistance and mechanical properties, the application and performance improvement of nickel-based alloys has been a popular research topic. For example, the Ni-Cr-Mo alloy is among a number of advanced nickel-based, corrosion-resistant alloys that are widely used, owing to its good corrosion resistance in most harsh corrosive environments, such as wet oxygen, sulfuric acid, strong oxidized salt and other media [6,7,8]. A feasible method is to prepare the Ni-based coating on the surface of the boiler tube for high-temperature corrosion protection. Laser-cladding technology can produce a nickel-based alloy cladding layer on the surface of the steel substrate with almost no porosity and good metallurgical bonding with the substrate, effectively avoiding t coating off and coating pores provide channels for corrosive media, resulting in substrate corrosion. Ni-Cr-Mo alloys have been widely studied as cladding materials due to their excellent corrosion resistance. Liu et al. [9] studied the thermal corrosion behavior of an Inconel 625 superalloy laser-cladding layer in a molten salt of Na_2_SO_4_-MgSO_4_ mixture at 900 °C. Li et al. [10] successfully prepared a Ni-Cr-Mo cladding layer on the TP347H stainless steel surface of a biomass boiler superheater tube using laser-cladding technology, and tested it by simulating the corrosion environment of the superheater surface of a biomass combustion device. It was found that the prepared cladding layer had excellent corrosion resistance. Liu et al. [11] compared the corrosion resistance of TP347H stainless steel, C22 alloy and a C22 alloy cladding layer prepared by laser cladding in molten chloride at 450–750 °C, and found that the corrosion resistance of the C22 alloy cladding layer was about twice that of the C22 alloy and ten times that of the TP347H. Liu et al. [12] simulated the severe high-temperature corrosion environment of a pulverized coal boiler in a low-NOx combustion environment. The experimental results showed that the corrosion resistance of the prepared C22 laser-cladding layer was not only significantly better than that of 20 g steel, but also higher than that of a commercial C22 alloy of similar composition.

In this work, Ni-Cr-Mo corrosion-resistant cladding layers with different Si contents (Si contents of 0 wt.%, 1 wt.%, 3 wt.% and 5 wt.%, respectively) were prepared on a Q235 steel surface by laser-cladding technology. Q235 steel is a common carbon structural steel with good plastic weldability and machinability. Being cheap and easy to produce, Q235 steel was very suitable for the substrate material. The phase composition and microstructure of the cladding layer were characterized, and the effect of Si on a Ni-Cr-Mo corrosion resistant alloy in a high-temperature environment with synergistic corrosion by S and Cl was investigated, which provided a theoretical basis for solving the corrosion problems of high-temperature molten salt in the boilers of waste-incineration power plants, and for the development of high-temperature corrosion-resistant materials.

## 2. Materials and Methods

### 2.1. Materials

The Ni-Cr-Mo laser-cladding layers with different Si contents were prepared on Q235 steel. The chemical composition of Q235 steel is shown in Table 1. Si powder was added into the Ni-Cr-Mo alloy powder provided by Beijing New Zhulian New Material Technology Co., LTD (Beijing, China)., and the mixed powders with mass fractions of Si of 0 wt.%, 1 wt.%, 3 wt.% and 5 wt.% were prepared by constant stirring for 10 h. It is worth noting that in order to avoid the influence of Cr and Mo content changes on the high-temperature corrosion resistance of the sample, in the process of preparing the mixed powder, Cr and Mo powders were added to keep the content of Cr and Mo fixed. The four mixed powders were named S0, S1, S3 and S5 in accordance with the respective Si contents (Si contents were 0 wt.%, 1 wt.%, 3 wt.% and 5 wt.% respectively), in order to facilitate the subsequent high-temperature molten chloride corrosion test, microstructure analysis and corrosion product analysis. The chemical compositions of mixed the powders are shown in Table 2.

### 2.2. Laser-Cladding Process and Preparation of Samples

The Ni-Cr-Mo cladding layers with different Si contents were prepared by laser-cladding technology. The working parameters were as follows: laser power, 2400 W; scan speed, 9 mm/s; single clad thickness, 700 μm. Argon was used to protect the molten pool during laser cladding. On the basis of single-channel cladding, a cladding layer with a thickness of about 4 mm was prepared by multi-channel lap bonding and multiple cladding. Then, the cladding layers with four components were processed to a size of 20 mm × 10 mm × 2 mm by wire cutting (the middle and upper parts of the cladding layer were selected as sampling locations to eliminate the diluting effect of matrix elements on the cladding layer). The cladding layer samples cut for the high-temperature corrosion test were polished by 400#, 600# and 800# waterproof sandpaper, successively, on the metallographic grinding machine until the surface of the cladding layer samples were smooth. The polished samples were ultrasonically cleaned in anhydrous ethanol solution for 10 min with ultrasonic cleaning instruments, and were then ultrasonically cleaned in acetone solution for 10 min to thoroughly remove the oil and impurities on the surface of the samples. After that, the samples were blown dry with a hair dryer and put into a drying dish for later use.

### 2.3. High-Temperature Corrosion Experiment

The incineration environment of a waste power plant is extremely complex, and there is no relatively fixed standard for corrosive media. In this work, the mixed molten salt corrosion reagent consisting of NaCl, KCl, Na_2_SO_4_ and K_2_SO_4_ was proportioned according to the mass fraction of 1:1:1:1, and the high-temperature corrosion experiment was carried out at 550 °C to simulate the high-temperature corrosion environment in the garbage generator set. Because the corrosion products were peeled off easily during the experiment, this experiment chose to measure the weight-loss-per-unit area of samples after corrosion to characterize the corrosion resistance of different samples. The calculation formula is as follows:(1)$$m=\frac{m_{0}-m_{1}}{S}$$
where *m*_0_ is the initial mass (g) of the experimental sample, *m*_1_ is the remaining sample mass (g) after completing the corrosion test and removing the corrosion products on the sample surface, and *S* is the surface area (m^2^) of the experimental sample in contact with the corrosive environment.

During the high-temperature corrosion test, the bottom of the small corundum crucible boat was first covered with corrosion reagents to a thickness of about 3 mm, then the sample was placed in the crucible boat, and the remaining space of the crucible boat was filled with corrosion reagent, ensuring that all 6 surfaces of the sample were covered by corrosion reagent. The small crucible boat was then placed in the large corundum crucible boat to facilitate the sample from the resistance furnace. The corrosion cycle was set to 24 h, a total of 7 cycles (168 h). The samples were used in a non-cyclic way; that is, a total of eight samples were placed in the tubular furnace at the beginning, and one sample was removed for acid pickling and weighing every 24 h during the corrosion process. The eighth sample was continuously corroded for 168 h. After being take out, it was not subjected to pickling and was weighed for subsequent XRD and SEM analysis of the corrosion mechanism. The above corrosion tests were repeated 3 times to obtain more accurate corrosion weight-loss data. The quality of the samples was measured with an electronic balance (accuracy: ±0.01 mg). The pickling processes were as follows: after complete cooling, the sample was removed from the corrosion reagent and the corrosion products were ultrasonically cleaned from the surface using deionized water, accelerating the stripping of corrosion products and corrosion reagents from the surface. The sample was subsequently pickled in 25 wt.% hydrochloric acid (concentration: 8.082 mol/L) at 80 °C to remove any corrosion products that had not been stripped [13].

### 2.4. Characterization Method

A scanning electron microscope (SEM) was used to observe the surface morphology of the laser-cladding layer (using aqua regia erosion) and the surface-corrosion products and sectional morphology of the cladding layer after corrosion. The composition of the corrosion products was measured by X-ray energy dispersive spectroscopy (EDS, Bruker, Billerica, MA, USA). The cross section of the sample was sealed with resin, ground, polished, and observed in a backscattering pattern. The phases of the corrosion products were analyzed by X-ray diffractometer (XRD). The schematic of the experimental part of this work is displayed in Figure 1.

## 3. Results and Discussion

### 3.1. Microstructure of the Cladding Layer

The XRD diffraction patterns of the four samples with different Si contents are shown in Figure 2. The main phases of the four cladding layers were γ-Ni solid solution, and the peak position of the cladding layer was slightly smaller than that of pure γ-Ni. The reason for the above phenomenon is that the solidification speed of molten pool was rapid during laser cladding, and most solute atoms, such as Cr, Mo and Si, were bound in the face-centered cubic lattice of γ-Ni. According to the Bragg equation (2dsinθ = nλ), the crystal plane spacing of the cladding layer was relatively large, which was mainly due to lattice distortion caused by the solution-strengthening effect.

Figure 3 clearly reveals the microstructure morphologies of the cladding layers with four different Si contents. It can be seen from the figure that the four cladding layers were mainly composed of two phases, namely the gray–black phase and the grid-like gray–white phase. At the same time, there were a few black inclusions and holes in the tissue. However, only peaks of the γ-Ni solid solution can be observed in the XRD diagram (Figure 1). EDS analysis was performed on the two phase components of each cladding layer sample, and the results are shown in Table 3. The element contents between the two phases are similar, and the difference mainly focuses on Mo, Si and Ni. The gray–white phase contains more Mo and Si, but less Ni. It can be inferred that both are produced by solid solution of Cr, Mo and Si into the γ-Ni matrix. The above phenomenon occurred because laser cladding is a process of rapid heating and cooling with high heat transfer and solidification rates, which prevents the full diffusion of Cr, Mo, Si and other elements, resulting in interdendrite segregation. However, the atomic radii of Mo and Si are larger than that of Cr, so the solid solution of them into γ-Ni requires more energy, resulting in greater lattice distortion. A large amount of Mo and Si could not be solidly dissolved into the lattice of γ-Ni, resulting in enrichment in the gray–white phase between dendrites. Along with the Si content increase, the differences in Ni, Mo and Si contents in the two phases were more significant. In addition, the microstructure of the four samples was similar (grid microstructure), but there were some differences. The size of the gray–white phase between dendrites is largest in the sample without Si (S0). With the increase in Si content, the grid-like gray–white phase size decreased gradually.

### 3.2. High-Temperature Corrosion Kinetics

Figure 4a shows the weight loss curves of the four cladding layers with different Si contents (S0, S1, S3 and S5) corroded for 168 h in the mixed salt corrosion reagent at 550 °C. It can be seen that the weight loss difference of the four cladding layers was not obvious at the initial stage of corrosion (within 48 h), while the four cladding layers showed maximum weight loss after 168 h. The maximum weight loss of S0 was 78.82 mg/cm^2^, that of S1 was 44.91 mg/cm^2^, that of S3 was 37.15 mg/cm^2^ and that of S5 was 30.11 mg/cm^2^. With the increase in Si content, the corrosion rate of the cladding layer tended to decrease. The corrosion resistance of S5 cladding with 5 wt.% Si was the best, with corrosion loss weight of only 38.2% of that of the cladding without Si. In addition, according to the actual weight loss curves, weight loss data of the four cladding layers can be fitted to predict the corrosion weight loss trend (the formulas obtained by fitting are shown in Table 4). The weight loss rate can be obtained by substituting the corrosion time into the derivative function. In the last period (168 h), the change rate of corrosion weight loss of the four cladding layers (unit: mg·cm^−2^·h^−1^) from high to low was S0 (0.68) > S1 (0.34) > S3 (0.29) > S5 (0.20), and that of S0 was 3.4 times that of S5. Combined with the weight loss curves, it can be seen that the corrosion weight loss rate of the three cladding layers with Si added was stable in the late corrosion stage, while that of the cladding layer without Si increased significantly. This indicates that with the progress of corrosion, the oxidation layer on the surface of S0 could no longer hinder the interaction between O, S, Cl and the surface metal, thus accelerating the occurrence of corrosion. Figure 4b shows the macroscopic morphology of the four samples after 168 h of corrosion at 550 °C. At 550 °C, green corrosion products (Cr-rich corrosion products, which are discussed below) were found on cladding samples of all four Si contents.

### 3.3. High-Temperature Corrosion Products

Figure 5 shows the XRD patterns of the corrosion products of the four cladding layers after corrosion at 550 °C for 168 h. It can be seen that Cr_2_O_3_ formed on the surface of all the four cladding layers at 550 °C. The oxygen in the air was adsorbed on the surface of the sample by diffusion, resulting inmetal oxide layers. The oxidation sequences of different elements in the Ni-Cr-Mo alloy vary, depending on the Gibbs free energy at the experimental temperature. In general, the Gibbs free energy of Cr_2_O_3_ formation is the lowest, followed by that of MoO_2_ and NiO. At 550 °C, only partial melting occurred in the corrosive medium (mixed salt), and chemical corrosion was still the main corrosion mechanism. O_2_ in the airfreely reacted with the cladding layer through the unmelted corrosive medium. The chemical reaction equation is as follows:xM(s) + (y/2)O_2_(g)→M_x_O_y_(s)(2)

In addition, the cladding layer could also react with sulfates in the corrosive medium, as follows:xM(s) + ySO_4_^2−^(g,l)→M_x_O_y_(s) + ySO_2_ + yO^2−^(3)

M is metal element (Ni, Cr, Mo and other metal elements), will generate NiO, Cr_2_O_3_, MoO_2_ and other oxides, which are attached to the surface of the cladding layer to prevent the continuation of corrosion. In addition, due to the presence of alkali metal chloride in the corrosion medium, the following reactions occur in the corrosion process:RCl(s)→RCl(g)(4)
RCl(s,g) + H_2_O(g)→ROH(s,g) + HCl(g)(5)
4HCl(g) + O_2_(g)→2Cl_2_(g) + 2H_2_O(g)(6)
where R is the reaction of Cl through the oxide layer on the surface and the cladding layer at high temperature to generate chloride. As confirmed by Abels et al. [14], this part of chloride (G) reacts with O_2_ again to generate corresponding oxide and Cl_2_. According to the reaction (6), Cl_2_ is the main corrosive medium in the short period of corrosion in the HCl-O_2_ atmosphere, rather than HCl. Therefore, the mechanism of Cl^−^- and Cl^−^-containing gases corroding the alloy is as follows [15,16,17]:M(s) + Cl_2_(g)→MCl_2_(s)(7)
M(s) + 2HCl(g)→MCl(s) + H_2_(g)(8)
MCl(s)→MCl(g)(9)
xMCl_2_(g) + (y/2)O_2_(g)→M_x_O_y_(s) + xCl_2_(g)(10)

Therefore, a large number of metal oxides (NiO, Cr_2_O_3_, etc.) were detected in the outer corrosion product layer. Meanwhile, with the increase in the content of Si, SiO_2_ was also detected in cladding-layer surface-corrosion products. SiO_2_ can also form a dense oxide film that effectively protects the alloy cladding layer during high-temperature corrosion. NiCr_2_O_4_ and CrMoO_3_ were also detected on the surface of some samples. Due to the higher elemental diffusion energy at higher temperatures, some Ni atoms replace the positions of Cr and Mo in the oxide layers. In other words, Cr_2_O_3_ and MoO_2_ underwent high-temperature melt, which caused the weight loss in the samples.

Figure 6 shows the surface-corrosion product morphology of the four cladding layers with different Si contents after continuous corrosion at 550 °C for 168 h. The surface of the four samples all have different degrees of corrosion pits. The inside and outside of the corrosion pits of the four samples were analyzed respectively. The EDS analysis results of the area around the corrosion pit of S0 and S1 indicate that the Ni and O contents of outer pits were high; however, the Cr content increased significantly inside the pits and was much higher than the Ni content. O content was very high, while the S content was noticeably lower than that outside of the pit. In addition, the corrosion products with obvious layer structures could be observed by analyzing the magnified morphology of the sample surface. The corrosion products in the pits and the upper-layer corrosion products outside of the pits contained both Cr and O, of which the content was much higher than other elements. The dense Cr_2_O_3_ film is beneficial to prevent the diffusion of O and corrosion reagents into the substrate, delaying the occurrence of corrosion. It can be deduced that when the Si content was too low, owing to the melting and volatilization of some Cr/Mo-rich oxides in the outermost corrosion products, the surface products could not block the action of corrosive gases and corrosive ions, and the internal alloy materials were constantly suffering from corrosion. However, Nickel oxide is more difficult to volatilize or decompose, because it possesses high stability compared to Cr and Mo compounds. Therefore, corrosion products that were poor in Cr and Mo and rich in Ni formed on the samples surface. With the increases in added silicon, the Si content in the Cr-rich oxides of the S3 and S5 also increased. The corrosion resistance of the superalloys is mainly due to the protective oxide film generated on their surfaces during high-temperature oxidation. The continuous oxide layer effectively prevents the intrusion of corrosive media, such as O, S and Cl, consequently resisting high-temperature oxidation and high-temperature thermal corrosion. It has been confirmed that the addition of Si to the alloy can promote the generation of dense Cr_2_O_3_ on the alloy surface, thus improving corrosion resistance [18].

To further explore the corrosion of Ni-Cr-Mo cladding layers with different Si contents, the cross-sectional morphologies of the four samples were observed after corrosion at 550 °C for 168 h (Figure 7). The morphologies of the outermost-layer corrosion products of the four samples were different. The outermost corrosion products on the surfaces without Si were loose and porous, while after Si was added, the outermost corrosion products on the sample surfaces changed and formed a denser film mainly composed of O and Cr. Moreover, the Cr-rich oxide scale on the surfaces of the samples with 3% and 5% Si was denser than on that with 1% Si. In some studies [19,20], electrochemical methods have been used to confirm that in molten salt environments, different states of Cr-rich oxide scale produce different degrees of protection, which is also the reason why the corrosion resistance of the cladding layer with Si addition was better than that without Si addition in this work. However, the higher carrier density of Cr-rich oxide layers in more intensely corrosive environments was also demonstrated by electrochemical measurements (the higher the carrier density, the more point defects, the more micropores, the rougher the surface, the weaker the protection effect of the corrosion layer). Combined with the morphology characteristics of the outer Cr-rich oxide layers of the four cladding layers in Figure 7, this can better explain why the corrosion resistance of S3 and S5 was better than that of S1. In addition, except for Si, the contents of other elements (Ni, Mo, etc.) at this position decreased significantly; that is, Si is also present in Cr-rich oxide positions. The addition of Si to Ni-based alloys contributes to the formation of chromium oxides at high temperature, which has been confirmed in previous reports [18,21]. The Cr-dissipative zone was observed in the cross-sections of the four cladding layers, revealing that Cr in the cladding layer near the corrosion layer is used to form chromium oxide film on the surface, and the width of the Cr-dissipative zone reduced noticeably after the addition of Si. However, it is noteworthy that the Cr-rich oxide scale is not observed in the sectional distribution diagram of the elements in the S0 sample without Si, which may have resulted from the destruction of the Cr-rich oxide scale generated previously, and the high-temperature corrosion resistance of the cladding layer that mainly comes from the rapid formation of a Cr_2_O_3_ oxide film to hinder the continuation of the reaction. The selective oxidation of the surface occurred firstly in high-temperature thermal corrosion, and the Cr-rich oxide scale was destroyed. O, S and Cl diffused into the cladding layer due to the destruction of the Cr-rich oxide scale, resulting in severely accelerated corrosion. When the surface oxide scale was mainly composed of NiO, oxidation corrosion was aggravated as the acid melting and fusing reaction of NiO and Na_2_SO_4_ occurred, so that the weight loss of the Ni-Cr-Mo cladding layer without Si in thermal corrosion was much greater than that of the Ni-Cr-Mo cladding layer with Si added, and increased significantly at later stages.

## 4. Conclusions

The main phases of the four Ni-Cr-Mo cladding layers with different Si contents were γ-Ni solid solutions comprising a primary crystal phase and an interdendritic eutectic phase. Si was mainly distributed in the interdendritic eutectic phase. The eutectic phase size decreased with the increase in Si content.

Si is improved the corrosion resistance of the cladding layers at high temperature. In NaCl-KCl-Na_2_SO_4_-K_2_SO_4_ mixed salt medium at 550 °C, the corrosion resistance of the cladding layers was improved with increase in Si content (Si content was between 0 and 5 wt.%). The corrosion weight loss of the Ni-Cr-Mo cladding layer with Si added was significantly less than that of the cladding layer without Si.

Si contributed to the formation of a dense oxide film of chromium on the on the surface of cladding layer. From the cross-section of the corrosion sample, the Cr-rich oxide layer does not appear in the outermost measurement of the Ni-Cr-Mo cladding layer without Si, but is very obvious on the surface of the cladding layer with Si. Moreover, the Cr-rich oxide scales on the cladding layers with 3 wt.% and 5 wt.% Si were denser than that on the cladding layer with 1 wt.% Si.

## Figures and Tables

**Figure 1 materials-15-03152-f001:**
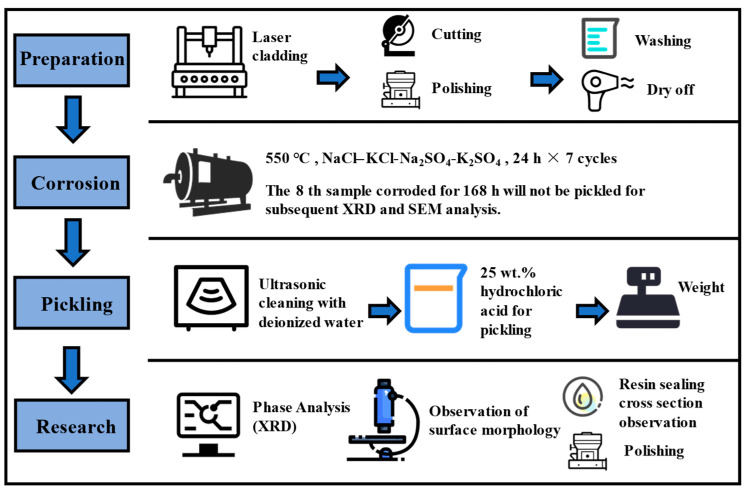
The schematic of the high-temperature corrosion experiment.

**Figure 2 materials-15-03152-f002:**
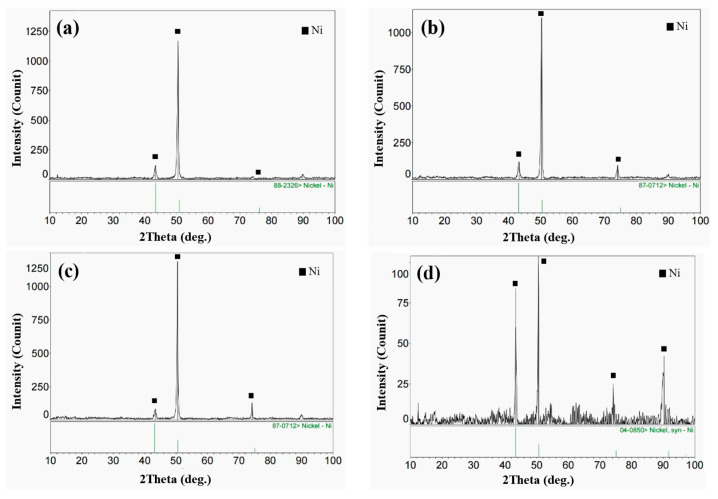
XRD patterns of the four cladding layers with different Si contents: (**a**) S0; (**b**) S1; (**c**) S3; (**d**) S5.

**Figure 3 materials-15-03152-f003:**
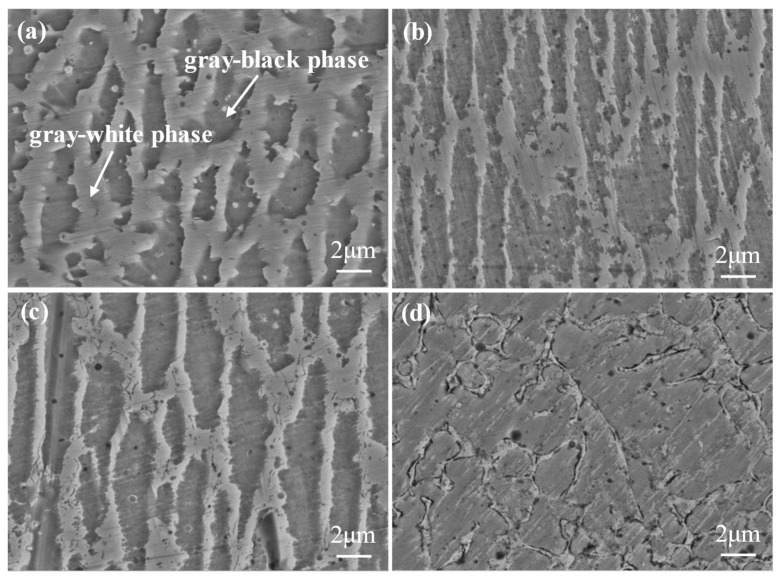
Microstructures of cladding layers with different Si contents: (**a**) S0; (**b**) S1; (**c**) S3; (**d**) S5.

**Figure 4 materials-15-03152-f004:**
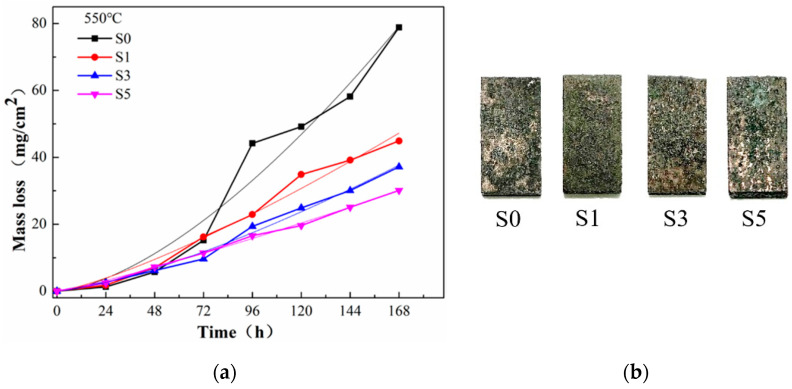
(**a**) The weight loss curves of the four cladding layers corroded for 168 h in the mixed salt corrosion reagent; (**b**) the macroscopic corrosion morphology of the four samples after 168 h.

**Figure 5 materials-15-03152-f005:**
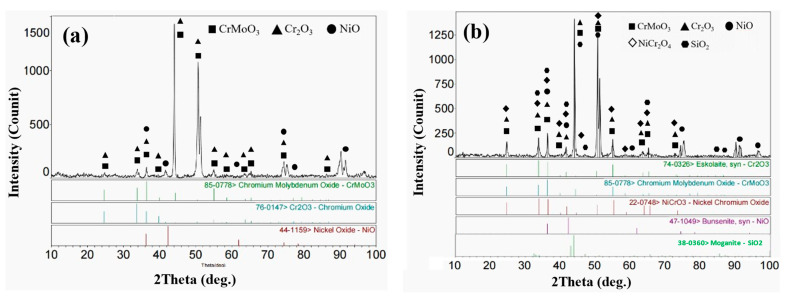
The XRD patterns of the corrosion products of the four cladding layers after corrosion for 168 h: (**a**) S0; (**b**) S1; (**c**) S3; (**d**) S5.

**Figure 6 materials-15-03152-f006:**
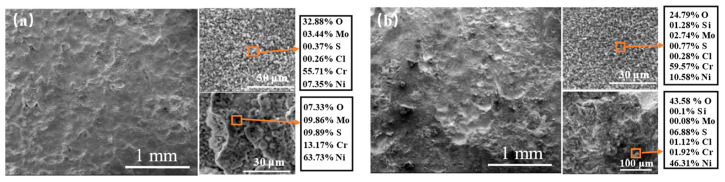
The surface-corrosion product morphology after etched at 550 °C for 168 h: (**a**) S0; (**b**) S1; (**c**) S3; (**d**) S5.

**Figure 7 materials-15-03152-f007:**
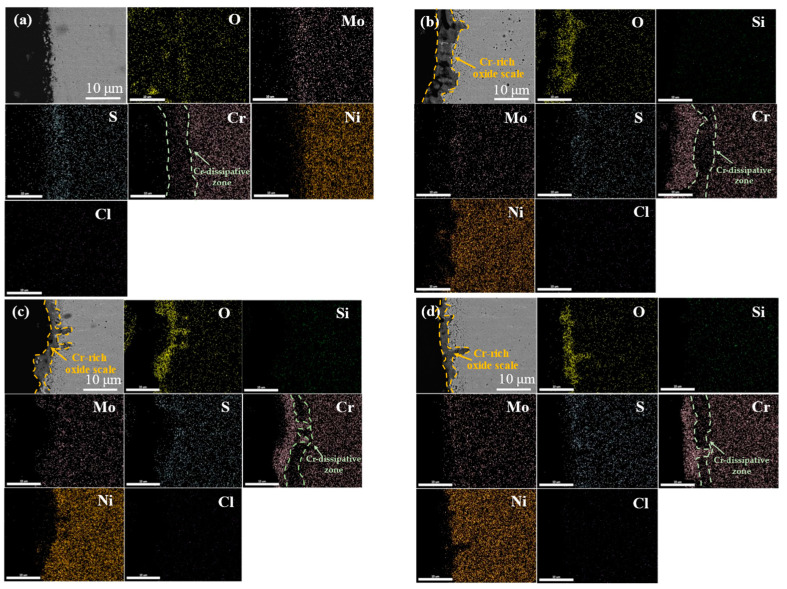
The cross-sectional morphology and EDS element distribution of cladding layer after high temperature exposure: (**a**) S0; (**b**) S1; (**c**) S3; (**d**) S5.

**Table 1 materials-15-03152-t001:** The chemical compositions of Q235 steel in wt.%.

Substrate	Element Content (wt.%)				
	Fe	C	Cr	Mn	Si
Q235	Bal	0.10–0.22	≤0.25	0.40–0.60	0.12–0.30

**Table 2 materials-15-03152-t002:** The chemical compositions of mixed powders in wt.%.

Samples	Element Content (wt.%)			
	Ni	Cr	Mo	Si
S0	Bal	24	13	0
S1	Bal	24	13	1
S3	Bal	24	13	3
S5	Bal	24	13	5

**Table 3 materials-15-03152-t003:** Chemical composition (wt.%) of two phases in four cladding layers with different Si contents.

Sample Number and Location		Element Content (wt.%)			
		Ni	Cr	Mo	Si
S0	gray–black phase	61.38	24.44	14.18	00.00
	gray–white phase	59.72	23.98	16.30	00.00
S1	gray–black phase	61.68	23.36	13.61	01.35
	gray–white phase	56.61	23.78	17.85	01.76
S3	gray–black phase	61.09	23.61	13.37	01.93
	gray–white phase	56.70	22.48	17.47	03.35
S5	gray–black phase	61.54	23.29	12.42	02.75
	gray–white phase	49.35	20.64	23.78	06.22

**Table 4 materials-15-03152-t004:** High-temperature kinetic curve-fitting curve parameters.

Sample	Equation	Anastomosis Degree	Derived Function	The Rate of Corrosion Change (168 h)
S0	y = 0.027x^1.558^	0.94909	y = 0.042x^0.558^	0.68 mg·cm^−2^·h^−^^1^
S1	y = 0.067x^1.279^	0.97504	y = 0.085x^0.279^	0.34 mg·cm^−2^·h^−1^
S3	y = 0.033x^1.372^	0.98796	y = 0.045x^0.372^	0.29 mg·cm^−2^·h^−1^
S5	y = 0.082x^1.152^	0.99618	y = 0.945x^0.152^	0.20 mg·cm^−2^·h^−1^

## Data Availability

Not applicable.

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
