# Peer review of "High-Temperature Corrosion of Ni-Cr-Mo Cladding Layers with Different Si Contents in NaCl-KCl-Na2SO4-K2SO4 Mixed Salt Medium"

_materials, 2022, doi:10.3390/ma15093152_

Round 1
Reviewer 1 Report
Review of paper no. materials-1689613 titled High temperature corrosion of Ni-Cr-Mo cladding layer with different Si contents in NaCl-KCl-Na2SO4-KSO4 mixed salt medium by S. Chen et al.
This is an interesting paper that studies the kinetics and mechanism of molten salt corrosion of a Ni-Cr-Mo cladding layer with different Si concentrations. It is found that the corrosion resistance increases with increasing Si content. The paper is publishable subject to revision.
1.It would help to provide a schematic of the high temperature corrosion experiment.
2.The authors say that they observed both primary phase and eutectic phase in the alloy microstructure (Fig. 2). However, only one phase - γ-Ni – is presented in XRD patterns (Fig. 1). What do you mean by the “eutectic phase”?
3.Why were the cladded alloys etched with strongly corrosive aqua regia prior to SEM observation (line 137)? It might have influenced the microstructure observed.
4.The oxidized samples were studied before and after pickling (lines 128 - 135). Which microstructure is presented in Fig. 6? Is it before or after pickling?
5.It seems that the oxides were not completely removed by etching/pickling (Fig. 5). It might have influenced the weight loss data (Fig. 3). Could you, please, compare the cross sections of the same alloy before and after pickling?
6.The concentration of 5 wt.% Si is usually sufficient to form a complete SiO2 layer. The complete silica layer can significantly hinder the corrosion. The silica layer is usually found under the chromia scale as SiO2 is more stable than Cr2O3. Why is there is no silica layer observed in the present case (Fig. 6)?
7.The XRD patterns (Figs. 1 and 4) should include the powder diffraction file numbers (PDF nos.) of the phases.
8.The readability of Figs. 1, 4, and 6 needs to be improved.
9.The English of the paper should also be improved.
Reviewer 2 Report
The presented manuscript includes the study of the High temperature corrosion of Ni-Cr-Mo cladding layer with different Si contents in NaCl-KCl-Na2SO4-K2SO4 mixed salt medium.
The results of the work are presented on a good level, but, some general corrections and weak points should be mentioned.
- Lines 151-152. Please prove written statements based on calculations that are not presented.
- How many samples were done in parallel? Please add SD everywhere where it is applicable (Table 2, Fig 3a).
- A lot of questions about the shown differences in XRD analysis and indicated phases. XRD diffraction patterns look absolutely similar. In similar peaks, the authors have shown different phases. Please elaborate on this part.
- Lines 244-246. Authors stated about pits analysis, speculate about pits depth without any cross-section, etc.
- Obviously, without electrochemical measurements paper looks a little bit simple. Please, compare your results with the published similar ones, like in your salts system for other alloys.
Reviewer 3 Report
The manuscript presents an interesting study about the Ni-Cr-Mo-Si coating deposited on the Q235 steel surface in order to improve its high-temperature corrosion resistance. The coating characteristics were studied by XRD, SEM etc. The paper needs minor revisions before it is processed further, some comments follow:
Abstract
Add a brief description of the main methods used to obtain and characterize the coating.
Introduction section
Add in the last paragraph and also in the abstract the material used as substrate.
Materials and Methods
Divide this section into three subsections, for example 2.1. Materials (Please introduce a table with the chemical composition of the used steel). 2.2. Laser cladding process and 2.3. Methods used to characterize the coating.
Introduce the pH value of the corrosive solution.
Results and Discussion
Figures 1, 3, 4, 5 and 6 are unclear. Please replace them.
Table 2. In order for the chemical composition to be as accurate as possible, EDS determinations must be performed in several areas of the sample and an average of the results calculated. Add the number of the measurements.
Future recommendation: It will be interesting if the corrosion behaviour will be analyzed by electrochemical impedance spectroscopy in different corrosive media.
Round 2
Reviewer 1 Report
Authors answered most of my comments. The paper can be accepted for publication.
Reviewer 2 Report
main points were corrected